# Treatment of Peripartum Depression with Antidepressants and Other Psychotropic Medications: A Synthesis of Clinical Practice Guidelines in Europe

**DOI:** 10.3390/ijerph19041973

**Published:** 2022-02-10

**Authors:** Sarah Kittel-Schneider, Ethel Felice, Rachel Buhagiar, Mijke Lambregtse-van den Berg, Claire A. Wilson, Visnja Banjac Baljak, Katarina Savic Vujovic, Branislava Medic, Ana Opankovic, Ana Fonseca, Angela Lupattelli

**Affiliations:** 1Department of Psychiatry, Psychotherapy and Psychosomatics, University Hospital of Würzburg, D-97080 Würzburg, Germany; Kittel_S@ukw.de; 2Department of Psychiatry, Faculty of Medicine & Surgery Msida, University of Malta, 2080 Majjistral, Malta; ethelfelice@gmail.com; 3Health-Mental Health Services, 2080 Majjistral, Malta; rachel.buhagiar@gov.mt; 4Departments of Psychiatry and Child & Adolescent Psychiatry, Erasmus University Medical Center, 3015 GD Rotterdam, The Netherlands; mijke.vandenberg@erasmusmc.nl; 5Section of Women’s Mental Health, King’s College London, London SE5 8AF, UK; claire.1.wilson@kcl.ac.uk; 6South London and Maudsley NHS Foundation Trust, Bethlem Royal Hospital Monks Orchard Road, Beckenham BR3 3BX, UK; 7Clinic of Psychiatry, University Clinical Center of the Republic of Srpska, 78000 Banjaluka, Bosnia and Herzegovina; visnjab76@hotmail.com; 8Department of Pharmacology, Clinical Pharmacology and Toxicology, Faculty of Medicine, University of Belgrade, P.O. Box 38, 11129 Belgrade, Serbia; katarinasavicvujovic@gmail.com (K.S.V.); brankicamedic@gmail.com (B.M.); 9Clinic for Psychiatry, University Clinical Center, 11000 Belgrade, Serbia; ana.opankovic@gmail.com; 10Center for Research in Neuropsychology and Cognitive-Behavioral Intervention, University of Coimbra, 3000-115 Coimbra, Portugal; ana.fonseca77@gmail.com; 11PharmacoEpidemiology and Drug Safety Research Group, Department of Pharmacy, PharmaTox Strategic Research Initiative, Faculty of Mathematics and Natural Sciences, University of Oslo, 0316 Oslo, Norway

**Keywords:** clinical practice guideline, depression, anxiety, antidepressant, psychotropic medications, peripartum

## Abstract

This study examined (1) the availability and content of national CPGs for treatment of peripartum depression, including comorbid anxiety, with antidepressants and other psychotropics across Europe and (2) antidepressant and other psychotropic utilization data as an indicator of prescribers’ compliance to the guidelines. We conducted a search using Medline and the Guidelines International Network database, combined with direct e-mail contact with national Riseup-PPD COST ACTION members and researchers within psychiatry. Of the 48 European countries examined, we screened 41 records and included 14 of them for full-text evaluation. After exclusion of ineligible and duplicate records, we included 12 CPGs. Multiple CPGs recommend antidepressant initiation or continuation based on maternal disease severity, non-response to first-line non-pharmacological interventions, and after risk-benefit assessment. Advice on treatment of comorbid anxiety is largely missing or unspecific. Antidepressant dispensing data suggest general prescribers’ compliance with the preferred substances of the CPG, although country-specific differences were noted. To conclude, there is an urgent need for harmonized, up-to-date CPGs for pharmacological management of peripartum depression and comorbid anxiety in Europe. The recommendations need to be informed by the latest available evidence so that healthcare providers and women can make informed, evidence-based decisions about treatment choices.

## 1. Introduction

Peripartum or perinatal depression, which is depression arising in the period between the start of a pregnancy and the end of the first postpartum year, to use a broad definition, affects approximately one in eight women [1]. Peripartum and perinatal depression are used interchangeably, although the former term relates more specifically to the woman. The disorder often persists throughout the peripartum period, with as many as 47% of women with postnatal depression having experienced an antenatal episode [2]. In many cases, depression concurs with anxiety, and this adds a substantial mental health burden to the woman [3]. One recent study has proposed multiple subtypes of perinatal depression, which differ in terms of symptom dimension and time of onset [4]. Women may, therefore, need tailored treatment strategies, including pharmacotherapy, depending on their individual depression course, timing of onset, and prominent symptom typology.

Perinatal depression is associated with a spectrum of obstetric and long-term negative outcomes in the offspring [5,6], including possible adverse impacts on the mother-infant relationship [7,8]. It also substantially affects women’s well-being and functioning, and it can even lead to suicide [9]. In moderate to severe cases or after non-response to first-line psychotherapy, pharmacotherapy with antidepressants is often needed [10]. Pooled results from 40 cohort studies [11] indicate that selective serotonin reuptake inhibitors (SSRIs) are the most commonly used antidepressants, with a population prevalence of filled prescriptions ranging from 3.5% before pregnancy to 3.0% during gestation and 4.7% in the first year postpartum. Augmentation with antipsychotics or adjuvant pharmacotherapy with benzodiazepines or sedative antihistamines may be needed in some cases [10]. Nevertheless, pregnancy remains a major driver for discontinuation of antidepressants, and 49% of those individuals who chose to continue have low antidepressant adherence [12,13].

The decision-making process about antidepressant treatment during pregnancy or lactation is complex, as it involves weighing the possible risk of exposure in utero or in breast milk against the potential adverse effects of sub-optimally treated maternal peripartum depression to both the mother and child. Clinical practice guidelines (CPGs) for peripartum depression management may facilitate this decision-making process. However, many countries have not established CPGs for peripartum depression, and for those available, the recommendations are not always uniform [14]. In 2018, one systematic review evaluated the content of the available CPGs, and it was found that only four countries recommend continuation into pregnancy of a pre-existing antidepressant treatment [14]. This prior work extracted only recommendations from CPGs adhering to the quality criteria of the Appraisal of Guidelines for Research and Evaluation (AGREE) instrument. Thus, there are still knowledge gaps on current clinical practices from CPGs not meeting such quality criteria. Furthermore, the extent to which national CPGs are followed in relation to antidepressant and other psychotropic prescribing remains unknown.

Therefore, the aim of this review was to examine the availability of national CPGs for treatment of peripartum depression with antidepressants across Europe and review their content and recommendations for the pregnancy and postpartum periods. We further evaluate antidepressant utilization data in women during the perinatal period as an indicator of compliance to the guidelines. To shed additional light on mental disorder co-morbidity, we evaluated whether CPGs for peripartum depression provide guidance on psychopharmacological treatment for co-morbid anxiety, along with prescription fill data for other psychotropics (i.e., antipsychotics and benzodiazepines) and sedative antihistamines during pregnancy and postpartum.

## 2. Materials and Methods

### 2.1. Search and Selection Criteria for Clinical Practice Guidelines

We conducted an extensive search of CPGs for treatment of peripartum depression in 48 countries in Europe, including member countries of the European Union, Schengen states, and other European countries. San Marino and the Holy See, both located geographically in Italy, were not included, as the former follows guidelines in Italy and the latter was not relevant. We combined multiple search strategies. First, we searched the literature in the Medline database (via PubMed) from inception to 31 August 2021 using the free text terms “antidepressant, peripartum, perinatal, pregnancy, postpartum, antenatal period, prenatal period, postnatal period, depression, mental health, psychiatric” and applied the filter for guidelines only. Second, we searched the Guidelines International Network (GIN) database using the terms “depression, peripartum, perinatal, pregnancy” on 31 August 2021. Third, we contacted directly via email the national members of Riseup-PPD COST ACTION (CA18138–Research Innovation and Sustainable Pan-European Network in Peripartum Depression Disorder) with an inquiry about the existence of a CPG for peripartum depression in the country. Last, we contacted researchers within peripartum psychiatry in various countries. No exclusion criteria were employed based on language. In the searches in Medline and GIN, we did not include published CPGs from countries outside Europe. We did not restrict the search to CPGs meeting the quality criteria of the AGREE instrument, as we aimed to gather as much information as possible about current clinical practices. Case reports and animal studies were excluded. We excluded CPGs on depression or mental health in adults which did not cover or mention peripartum depression within them and CPGs on peripartum depression that did not mention pharmacotherapy interventions. Clinical recommendations without clear references or without a description of the process that led to the recommendation were also excluded. The literature searches and abstract screenings were performed by a single author. The selection of the CPGs eligible for inclusion were agreed upon by all authors.

Data abstraction was performed by one author depending on the relevant language and, thereafter, quality-checked by another author. We extracted recommendations regarding (1) initiation, continuation or discontinuation, and switching of the antidepressant for both new and preexisting depression in pregnancy or postpartum, (2) preferred and non-preferred antidepressants in pregnancy and while breastfeeding, (3) compatibility of antidepressants with breastfeeding, (4) antidepressant level monitoring or dose adjustment, and (5) recommendations for pharmacological treatment of comorbid anxiety in pregnancy and postpartum.

### 2.2. Search and Selection Criteria for Antidepressant and Psychotropic Utilization Studies

We searched the literature in the Medline database (via PubMed) from inception to 31 August 2021 using the free text terms “antidepressant, psychotropic, antipsychotic, anxiolytic, “medication use”, “drug use”, peripartum, perinatal, pregnancy, postpartum, antenatal period, prenatal period, postnatal period, depression, mental health, psychiatric”. We extracted the most complete or recent antidepressant drug utilization studies among those published in the last 10–15 years originating from countries in Europe. We applied no restriction as to the way antidepressant use in pregnancy and postpartum was measured in the studies (e.g., based on self-reporting, prescription fills, or medical records). The outcome criteria were prevalence estimates for antidepressant use before, during, and after pregnancy. The same criteria applied to the search and data extraction for other psychotropic medications. If available, we extracted prevalence estimates from more than one study.

### 2.3. Ethics Statement

No ethics approval was sought, as this review evaluated existing clinical practice guidelines. No informed consent was collected, as the study did not involve patients. The synthesis was not registered in PROSPERO.

## 3. Results

### 3.1. Identified Clinical Practice Guidelines

Figure 1 describes the flow diagram of the various search strategies to achieve the final sample of CPGs included in the study. Across the 48 countries examined, we were unable to identify a contact person or did not receive a response in 22 (45.8%) of the countries. We received a response or identified a CPG in the literature search for 26 countries in Europe, of which 10 (38.5%) (i.e., Austria, Bulgaria, Croatia, Cyprus, France, Greece, Iceland, Portugal, Turkey, and Bosnia and Herzegovina) did not have a national CPG for intervention strategies of peripartum depression or mental health, either specific or broader for the adult population, with mention of the peripartum population. In Ireland, we could only retrieve an information leaflet on peripartum depression for women, which is not classified as a CPG. In Ukraine (personal communication), the criteria for treatment of peripartum depression were reported to be in place, which included pharmacotherapy interventions with amitriptyline, phenazepam, relanium, frenolone, and with vitamins (e.g., ascorbic acid). However, no further information was obtained. Belgium and Sweden used protocols or guidelines for screening and treatment of peripartum depression based on international guidelines (NICE). However, pharmacotherapy interventions are not mentioned [15,16]. Of the searches in PubMed and the GIN database, we screened three CPGs from Spain, Poland, and the UK, which were duplicates of the ones obtained via the contact persons in these countries. We included and fully evaluated 12 CPGs. In the CPG from Latvia, recommendations on pharmacological interventions were only provided for the postpartum period. 

### 3.2. Pharmacological Interventions for Treatment of Antenatal Depression

Table 1 shows that most CPGs advise initiation of antidepressants in women with new onset or moderate-to-severe antenatal depression. This treatment should be undertaken after an individualized risk–benefit evaluation and following non-response to psychotherapy. In contrast, the CPG in Poland discourages the use of antidepressants in the first trimester and states that this medication should be discontinued before delivery. All the CPGs seem unanimous in recommending or mentioning the possibility to continue antidepressants in pregnancy for preexisting moderate-to-severe depression (Table 1). In the UK CPG, monotherapy (if possible) and the lowest effective dose are advised in the context of both initiation and continuation of the antidepressant. On the basis of filled prescription and drug utilization data, there was a decrease in the prevalence of antidepressant use from preconception (range: 1.6–9.6%) into pregnancy (range: 0.3–4.1%), with SSRI being the most commonly prescribed group in most countries (Table 1). For many countries in Eastern Europe, no such utilization data were available.

There is general agreement between the CPGs in evaluating individual drug response in the period prior to pregnancy in the decision making about antidepressant continuation during pregnancy. The CPGs provide less uniform guidance regarding switching antidepressants during pregnancy (Table 1). In the CPGs of Malta and Norway, switching is discouraged unless the drug is ineffective. The CPG in the Netherlands considers switching from paroxetine to a preferred antidepressant but before pregnancy. Multiple CPGs (Finland, Germany, Italy, and Serbia) do not provide guidance on switching. Likewise, information on antidepressant level monitoring in serum or plasma and on dosage adjustment is missing for the CPGs in Italy and Denmark.

Multiple CPGs mention sertraline and citalopram as preferred antidepressants in pregnancy, whereas others (i.e., Finland, Serbia, and Spain) list the class of SSRIs. Paroxetine was mentioned as not a preferred antidepressant in most CPGs, except for Serbia, the UK, and Norway. In the two latter countries, the CPGs advise basing the choice of the antidepressant on maternal prior response and its safety profile. Generally, the antidepressants recommended in the CPGs were also the ones most often used in gestation, except in Denmark (for fluoxetine), the Netherlands and Spain (for paroxetine), and Germany (for amitriptyline). Paroxetine ranked among the most commonly used antidepressants in pregnancy in specific countries (i.e., Italy, the Netherlands, or Spain).

### 3.3. Pharmacological Interventions for Treatment of Postpartum Depression

Table 2 summarizes the content and recommendations of the CPGs for the postpartum period. Most CPGs (*n* = 11) recommend initiation or continuation of antidepressant medications in women suffering from depression in the postpartum period. Nearly all CPGs suggest an individual risk–benefit evaluation of the antidepressant treatment in the case of breastfeeding. Recommendations about breastfeeding compatibility with maternal antidepressant use were not specified in three GPGs (Spain, Serbia, and Norway). The CPGs in the Netherlands, Italy, and Finland state that antidepressant use does not prevent breastfeeding, whereas the UK and Denmark advise closely monitoring the exposed breastfed infant for potential adverse effects, such as weight gain. The Maltese CPG advises that only healthy and full-term infants should be breastfed when mothers are taking antidepressants. The Polish CPG gives a detailed recommendation about the timing of antidepressant intake and breastfeeding (i.e., to take one daily dose before the longest sleep of the child and breastfeed directly before that). Recommendations about switching antidepressants are either unspecified (*n* = 5) in the CPG or discouraged, especially if it affects the woman. The prevalence estimates of antidepressant use postpartum were greater than in the antenatal period and generally returned to the magnitude seen pre-pregnancy. For most of the countries included in this work, no antidepressant utilization data postpartum are available.

The specific substances recommended and not recommended vary considerably between the CPGs, but taken together, sertraline (8/12 CPGs) and paroxetine (5/12 CPGs) are the ones most commonly preferred, while fluoxetine is not preferred in most CPGs (8/12 CPGs) due to its very long half-life with the risk of accumulation in the infant. Paroxetine, citalopram, sertraline, or SSRI in general are also the antidepressants most commonly taken by women postpartum. Fluoxetine does not rank high in drug utilization studies in the postpartum period.

### 3.4. Pharmacological Interventions for Antenatal or Postpartum Comorbid Anxiety and Use of Other Psychotropics

Treatment recommendations for comorbid anxiety are largely missing for both the antenatal and the postpartum period (Table 1 and Table 2). Only seven GPGs state that benzodiazepines can be offered in the case of severe anxiety during pregnancy but only for short-term treatment. In Malta, benzodiazepines are recommended only as needed, and the treatment of choice is augmentation with quetiapine, both during pregnancy and postpartum. During the latter period, sedative antihistamines represent a treatment option. In the UK, it is advised to treat comorbid anxiety with antidepressants during pregnancy or short-term benzodiazepines, and the latter medication is discouraged at postpartum in case of breastfeeding. The CPG in Latvia recommends treatment of comorbid anxiety postpartum with mirtazapine or atypical antipsychotics, including olanzapine at a low dose, while benzodiazepines should be avoided.

Prenatal use data for benzodiazepines, antipsychotics, and quetiapine specifically are lacking for some countries, and for sedative antihistamines, data are very sparse. During pregnancy, benzodiazepines are often used to a larger extent than antidepressants in specific countries (i.e., Germany, Poland, Serbia, and Spain), while in Norway, the use of benzodiazepine and sedative antihistamines is comparable (about 1%). With regard to the use of other psychotropic medication (as an add-on) in the postpartum period, utilization data are largely unavailable, as only the Nordic countries and the UK report postpartum use of benzodiazepines and antipsychotics in the ranges of 0.8–3.2% and 0.2–0.4%, respectively. 

## 4. Discussion

This review across European countries reports important gaps in the availability, agreement, and up-to-date evidence-based content of CPGs for the pharmacological treatment of peripartum depression. This may have implications in the decision making and uptake of effective treatment among perinatal women and consequently in reducing the pervasive costs of peripartum depression. Several of our findings are important for clinical practice and perinatal drug research at large. First, we identified a national CPG only in 12 out of the 48 countries in Europe, adding 6 guidelines to the latest synthesis by Molenaar et al. in 2018 [14]. Nevertheless, the absence of a CPG in most countries raises clear concerns about the pharmacological management of depression in pregnant women and new mothers [10], especially in countries where higher rates of peripartum depression [50,51] are paralleled by low use of antidepressants and greater use of benzodiazepines [24]. Second, we found general agreement within the CPGs in recommending psychotherapy as first-line intervention, as well as antidepressant initiation or continuation based on psychotherapy non-response or depression severity. However, the recommendations are sometimes unspecific and not uniform across guidelines. Third, emerging issues and questions that are met in the real-world practice are not covered with the latest available evidence (e.g., drug monitoring or dose adjustments, antidepressant switching and treatment augmentation, adjuvant strategies for comorbid anxiety, and compatibility of breastfeeding with antidepressant treatment). Finally, the unavailability of antidepressant and other psychotropic utilization data from pre-pregnancy through the end of the first postpartum year, especially in some countries, impedes the evaluation of prescribers’ compliance to a CPG and calls for ad hoc perinatal drug utilization research.

The available evidence about antidepressant safety and antidepressant effectiveness in the context of continuation, discontinuation, or initiation is limited, especially for the pregnancy period [50,52,53,54,55,56,57]. It is now widely acknowledged that intrauterine exposure to SSRIs does not substantially increase the risk of congenital anomalies in offspring, while the risk for negative longer-term developmental outcomes is less clear [58,59,60]. However, only more recent studies have compared outcomes in offspring born to continuers versus discontinuers [61]. Antidepressant continuation in pregnancy was found to increase the risk of low birth weight, premature birth, or affective disorder diagnosis later in childhood [62,63,64,65]. Yet, the role of confounding by maternal disease severity remains an important concern in this research. Regarding antidepressant effectiveness, a recent meta-analysis [53] found a 74% increased risk of depression relapse during pregnancy with antidepressant discontinuation relative to continuation in pregnancy. The four included studies were, however, very heterogeneous and adopted an oversimplified definition of antidepressant continuation or discontinuation that did not reflect the treatment intensity, dose changes, or timing of exposure as in real-world settings [66,67,68].

No observational or randomized study to date has investigated the benefit of antidepressant initiation in pregnancy on relapse or remission of peripartum depression. The need for clinical drug trials in pregnant and postpartum women has never been greater [69]. The findings from the “stop or go” randomized trial indicated no significant difference in the risk of relapse of depression in women who tapered SSRIs with additional preventive cognitive therapy, relative to those who continued SSRIs [70]. However, the study included only 44 women, demonstrating the need for larger trials which also address the efficacy of antidepressant initiation in pregnancy. In 2019, the Food and Drug Administration in the US approved the first drug specifically for the treatment of postpartum depression: the GABA-A receptor modulator brexanolone. Brexanolone is not yet approved in the EU, but regulatory pathways have been initiated for future marketing authorization. This new drug constitutes an important therapeutic option for women with severe postpartum depression, but its difficult administration in terms of duration and form (i.e., intravenously) may limit its usage. Determining the comparative effectiveness and safety of brexanolone versus any other treatment for postnatal depression [54] will be crucial to inform clinical decisions involving CPGs at an international level. Similarly, more research is needed about the comparative effectiveness of different pharmacological interventions versus other therapeutic options, such as electroconvulsive therapy for treatment of severe perinatal depression.

There remains a need for more unified guidelines on the use of antidepressants to treat peripartum depression to guide clinical decision making. However, the decision to treat peripartum depression with antidepressants must always consider the individualized risk–benefit profile of the medication for each woman [54]. Most CPGs recommend an individualized risk–benefit assessment, which should consider the psychiatric history of the woman, her response to prior or ongoing antidepressants, mental health outcomes following prior attempts to discontinue the medication, the woman’s treatment preference, and her desire to breastfeed. The antidepressant with the lowest known risk for breastfed children in the lowest effective dose and in the lowest effective drug serum concentration should be prescribed [60]. To the best of our knowledge, there is no evidence base to discourage breastfeeding of preterm or low birth weight infants. However, caution is needed due to the immature liver metabolic capacity in preterm infants, especially in combination with maternal fluoxetine use, which has a long half-life and increased risk of accumulation in the breastfed infant [60]. The decision making in pregnancy and while breastfeeding could be aided by further development of patient decision aid (PDA) tools. Early data suggest that they are acceptable to users and reduce decisional conflict [71,72].

Generally, there was satisfactory compliance in prescribing preferred antidepressants during pregnancy (e.g., sertraline and citalopram), although exceptions were noted. Paroxetine ranked among the most commonly used antidepressants in pregnancy in specific countries, despite being a non-preferred antidepressant. However, we could not corroborate whether this drug choice was derived from an individualized assessment based on maternal prior response to the drug or whether it reflects poor prescriber compliance to the CPG. One Dutch study [73] found that gynecologists and midwives were aware of the national CPG on antidepressants in pregnancy, yet only 13.9% of them adhered to its recommendations. Efforts are, therefore, necessary to facilitate the uptake of the CPG recommendations in routine clinical practice by all healthcare professionals involved in the care of women with peripartum depression.

One key finding is that guidance on intervention strategies for comorbid anxiety and advice on augmentation with antipsychotics are largely missing across the examined CPGs, and when present, it is too unspecific with regard to drug selection and maximum permissible doses. Indeed, we observed important country-specific fluctuations in the utilization of benzodiazepines that need to be addressed. Uniform, specific recommendations for this problem are needed for multiple reasons: (1) some women manifest active depressive symptoms despite antidepressant treatment, and clinicians need evidence-based guidance to treat them; (2) anxiety is a prominent symptom of severe peripartum depression [4]; and (3) benzodiazepines should be used only sporadically during pregnancy or postpartum, and alternative interventions are necessary for protracted treatments [60]. Yet, to date, evidence does not exist to help make recommendations for the perinatal population, which calls for urgent population-based perinatal drug research.

### Strengths and Limitations

Several strengths and limitations need mentioning. One of the main strengths of this synthesis is that we provided a global view of the existing CPGs across Europe. We applied multiple search strategies, our search was not restricted to CPGs meeting the AGREE instrument, and we applied no language restrictions, which enabled us to gather as many CPGs as possible, including current clinical practices. Direct contact with representatives of the COST network and experts in psychiatry and psychology allowed us to examine CPG availability in low- and middle-income countries in Europe, which are unlikely to publish national CPGs. Our review did not include consensus statements or expert opinion articles, as these items only reflect individuals’ perspectives or practices. In addition, we extracted psychotropic utilization data from the literature as a proxy of prescribers’ compliance to their national CPGs. However, such a proxy is not ideal, and specific field studies are necessary to accurately measure prescribers’ adherence to the CPGs [73]. Identification of CPGs eligible for inclusion in the review was performed by a single author, but the final decision for inclusion or exclusion was agreed upon by all authors. We did not assess the quality of the included CPGs based on the AGREE instrument, and therefore, we could not assess the degree of the evidence upon which the different CPG recommendations were based. Lastly, our review was restricted to European countries, and so our results are not generalizable to countries outside Europe.

## 5. Conclusions

Many countries in Europe do not have a CPG for pharmacological treatment of peripartum depression, and where present, recommendations are not fully uniform and not up to date with the latest available evidence. This review expresses the urgent need for a harmonized, up-to-date CPG for pharmacological management of peripartum depression and comorbid anxiety in Europe. Treatment recommendations need to be informed by the latest available evidence and cover emerging issues that are met in the current clinical practice. Our work is only the first step in facilitating the complex decision making in pharmacological treatment of women with peripartum depression. Women across Europe should be empowered to make informed, evidence-based decisions about their treatments during pregnancy and while breastfeeding.

## Figures and Tables

**Figure 1 ijerph-19-01973-f001:**
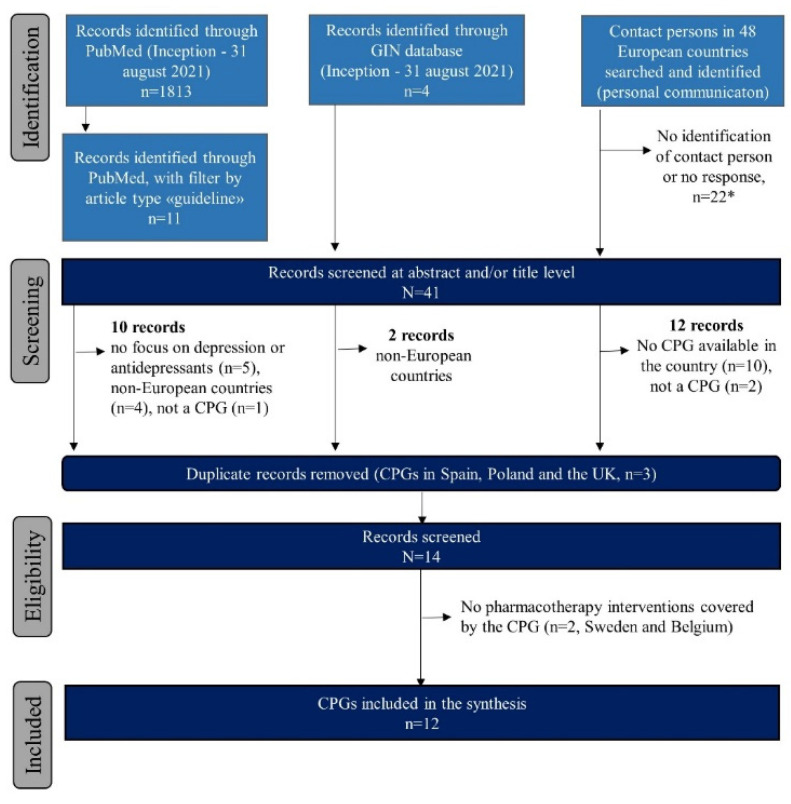
Flow chart of the review process for the clinical practice guideline synthesis. Abbreviations: CPG = clinical practice guideline; GIN = Guidelines International Network. * No response or identification in Albania, Andorra, Armenia, Azerbaijan, Belarus, Croatia, Czech Republic, Estonia, Georgia, Hungary, Kazakhstan, Kosovo, Liechtenstein, Luxembourg, Moldova, Montenegro, North Macedonia, Romania, Russia, Slovakia, Slovenia, and Switzerland.

**Table 1 ijerph-19-01973-t001:** Overview of recommendations in the CPGs about antidepressant treatment in women with antenatal depression, with prevalence estimates of antidepressant and other psychotropic medication use in the country.

Country, Publication Year, Type	New Depression, Initiate AD	Preexisting Depression, Continue AD	AD Dose Adjustment and Monitoring	Switching AD	Preferred or Not Preferred AD	AD Use before vs during Pregnancy (%)	Most Common ADs Used during Pregnancy	Treatment of Co-Morbid Anxiety	Other Psychotropics during Pregnancy (%)
Denmark [17] PMH-S	Yes, if severe and no response to psychotherapy	Yes	NS	Yes, if effective for woman’s depression	Sertraline, citalopram / Paroxetine, fluoxetine	2.0 vs. 1.9–4.1 [18,19]	Citalopram, sertraline, fluoxetine [19]	NS	BZD: 0.6 [20] AP: 0.4 [21] Quetiapine: 0.2 [21]
Finland [22] N-PPD	NS	Yes, in moderate-to-severe cases Psychotherapy first line	Monitoring of response is important	NS	SSRIs / Paroxetine, fluoxetine, tricyclics	1.6–4.0 [23] vs. 3.6* [24]	NA	NS	BZD: 1.2* [25] AP: 0.8 [21] Quetiapine: 0.9 [21]
Germany [26] N-PPD	Yes, after individual risk benefit evaluation, individual disease history, preference and availability of alternative treatments	Yes, in moderate-to-severe cases. Abrupt discontinuation is discouraged. Psychotherapy first line	Monotherapy if possible, lowest effective dose Continuous measurement of plasma levels	NS	Sertraline, citalopram / Paroxetine, fluoxetine	4.0 [27] vs. 0.4 (unpublished data)	Amitriptyline, (es)citalopram, sertraline (unpublished data)	NS	BZD: 3.3 [25] AP: 0.3 [21] Quetiapine: 0.2 [21]
Italy [28] PMH-S	Yes, after individual risk–benefit assessment	Yes, after individual risk–benefit assessment	NS	NS	NS	3.3–4.4 [29] vs. 1.2–1.6 [29]	Paroxetine, sertraline, citalopram [29]	Yes, BZD can be used	BZD: 1.4 [30] AP: 0.8 [31] Quetiapine: NA SAH: 0.4* [24]
Malta [32] PMH-S	Yes, after individual risk–benefit assessment; drug choice based on lowest risk, monotherapy if possible and at the lowest effective dose	Yes, after individual risk-benefit assessment; drug choice based on lowest risk, monotherapy if possible and at the lowest effective dose; previous response is considered	NS	If possible, switch paroxetine to other SSRI	Sertraline ¥; Fluoxetine / Paroxetine	NA	Sertraline, fluoxetine (unpublished data)	BZD only short term for extreme anxiety or agitation; BZD should be avoided in late pregnancy	BZD: NA AP: NA Quetiapine: NA
The Netherlands [33] PMH-S	Yes, after individual risk–benefit assessment	Yes, if woman is stable with medication	Yes, lowest effective dose; paroxetine preferably not >20 mg/day	If possible, switch paroxetine to other SSRI but pre-pregnancy	Sertraline ¥ / Paroxetine ¥	3.9 vs. 2.1 [34]	Sertraline, paroxetine, citalopram [34]	NS	BZD: 1.1 [25,35] AP: NA Quetiapine: NA
Norway [36] PMH-S	Yes, if severe and with non-pharmacological therapy	Yes, after individual risk–benefit assessment. Psychotherapy first line. Abrupt discontinuation is discouraged.	Yes, serum concentration	No	Choice based on prior drug response and its safety profile	2.0 vs. 1.5 [37]	(Es)citalopram, sertraline, venlafaxine [19]	NS	BZD: 0.9 [38] AP: 1.2 [21] Quetiapine: 0.3 [21] SAH: 1.0 [24] *
Poland [39] PMH-S	Individual risk–benefit assessment to be made. **AD in 1 trimester should be avoided, and AD should be discontinued before delivery	If severe depression or ongoing mild-to-moderate symptoms, AD should be considered. Gradual discontinuation if mild symptoms with psychotherapy. ** As for new depression.	Monotherapy, lowest effective dose	Yes, switching an AD which is effective and offers fewer adverse effects	NS / Paroxetine	- vs. 0.3 [24] *	NA	Yes, but do not offer BZD except for the short-term treatment of severe anxiety and agitation	BZD: 0.2–14.0 [24,25] * AP: NA Quetiapine: NA SAH: 0.4 [24] *
Serbia [40] N-PPD	Yes, if severe after individual risk–benefit assessment	Yes, after individual risk–benefit assessment	Yes, serum concentration	NS	Fluoxetine, citalopram, fluvoxamine, paroxetine, sertraline / TCA	- vs. 0.3 [24] *	(Es)citalopram, sertraline, mirtazapine, duloxetine (unpublished data)	Yes, BZD	BZD: 0.2–14.0 [24,25] * AP: NA Quetiapine: NA SAH: 0.4 [24] *
Spain [41] PMH-S	Yes, after individual risk–benefit assessment	Yes, after individual risk–benefit assessment and based on individual drug response	Monotherapy if possible, lowest effective dose; continuous measurement of plasma levels due to fluctuations in pregnancy is recommended	Yes, if lower risk to child and effective in mothers	SSRIs / Paroxetine, tricyclics, fluoxetine	- vs. 0.5–0.8 [42,43]	Paroxetine, citalopram, fluoxetine44	Yes, but for acute symptoms for maximum 4 weeks	BZD: 1.9 [42] AP: 0.1 [43] Quetiapine: NA
United Kingdom [44,45] PMH-S	Yes, particularly for moderate-to-severe depression, after discussing with the woman the risk–benefit assessment of AD; drug choice based on lowest risk, monotherapy if possible and at the lowest effective dose	Yes, particularly for moderate-to-severe depression, after discussing with the woman the risk–benefit assessment of AD; monotherapy if possible and at the lowest effective dose	Yes, dosages may need to be adjusted in pregnancy	Option to be discussed with the woman but aim is to expose fetus to as few drugs as possible	Unspecified, choice based on prior drug response and its safety profile	8.8–9.6 vs. 3.7 [29]	Fluoxetine, citalopram [29]	Yes, with ADs. Do not offer BZD except for the short-term treatment of severe anxiety and agitation.	BZD: 1.2 * [25] AP: 0.3–4.6 [21,46] Quetiapine: 0.4 [21]

AD = antidepressant; BZD = benzodiazepines; AP = antipsychotics; SAH = sedative antihistamines; PMH-S = peripartum mental health-specific; N-PPD = not specific to peripartum depression, but pregnant women are dealt with within the guideline for adult depression. Estimates of sedative antihistamines are only shown when available. As such, data are lacking for most countries. * Average estimate for the region at aggregated level (non-country specific) or for the specific country within the meta-analysis. ¥ Applies to first episode in pregnancy, when the woman starts on a new medication.

**Table 2 ijerph-19-01973-t002:** Overview of recommendations in the CPGs about antidepressant treatment in women with postnatal depression, with prevalence estimates of antidepressants and other psychotropic medication use in the country.

Country, Publication Year, Type	Depression, Initiate or Continue AD	AD Intake by Time of BF	Switching AD	Preferred or Not Preferred AD	AD Use Postpartum (%)	Most Common ADs Postpartum	Treatment Co-Morbid Anxiety	Other Psychotropic Postpartum (%)
Denmark [17] PMH-S	NS	No, but weight gain in infant should be monitored. Formula can be considered.	No, if the AD is effective and was taken in pregnancy	Sertraline, paroxetine / Fluoxetine, (es)citalopram, fluvoxamine, venlafaxine	4.1 [29]	NA	NS	BZD: 1.3 [20] AP: NA Quetiapine: NA
Finland [22] N-PPD	As for non-pregnant adults, psychotherapy is recommended for mild symptoms	No, use of SSRI does not prevent BF	NS	SSRIs / Fluoxetine	NA	NA	NS	BZD: 0.7–3.2 [47] AP: NA Quetiapine: NA
Germany [26] N-PPD	Yes, after risk–benefit analysis for mother and child and individual disease history, preference, and availability of alternative treatments	Yes, after risk–benefit analysis for mother and child	NS	SSRIs, tricyclics / NS	NA	NA	NS	BZD: NA AP: NA Quetiapine: NA
Italy [28] PMH-S	Yes, after risk–benefit analysis for mother and child	No, use of SSRI does not prevent BF	NS	NS / Fluoxetine	2.5–3.4 [29]	NA	Yes, short-term acting BZD	BZD: NA AP: NA Quetiapine: NA
Latvia [48] PMH-S	Yes, after risk–benefit analysis for mother and child in case of BF. For initiation of AD, start with lowest effective dose.	Assess whether dosage and regimen are compatible with BF	NS	SSRIs, sertraline / Fluoxetine	NA	NA	Yes, mirtazapine or atypical AP; quetiapine for augmentation therapy. Olanzapine only at low doses. BZD should be avoided.	BZD: NA AP: NA Quetiapine: NA
Malta [32] PMH-S	Yes, after individual risk–benefit assessment; drug choice based on lowest risk, monotherapy if possible and at the lowest effective dose	Yes, after individual risk–benefit assessment; drug choice based on lowest risk, monotherapy if possible, and at the lowest effective dose, previous response is considered	NS	Iimipramine, nortriptyline, sertraline / Citalopram, fluoxetine	NA	SSRIs e.g., sertraline, paroxetine (unpublished data)	Short-term BZD (caution in BF). Close monitoring of babies exposed to BZD via breastmilk. Diazepam should not be used while BF.	BZD: NA AP: NA Quetiapine: NA
The Netherlands 33 PMH-S	Yes, continue SSRI after delivery	Yes, BF is recommended	No, no evidence for switching	Paroxetine, sertraline / Fluoxetine, citalopram	3.1 [34]	Paroxetine, citalopram, sertraline [34]	NS	BZD: NA AP: NA Quetiapine: NA
Norway [36] PMH-S	Yes, if severe after individual risk–benefit assessment	NS	No	Sertraline, paroxetine / Doxepin, fluoxetine, citalopram	1.0 [37]	NA	NS	BZD: 0.8 [37] AP: 0.2 [37] Quetiapine: NA
Poland [39] PMH-S	Yes, initiate if severe and continue to prevent relapse. If history of severe depression or ongoing mild-to-moderate symptoms, AD should be considered.	Yes, AD in one daily dose before the child’s longest sleep, and BF is recommended just before AD intake	No, same treatment pattern should be used after delivery	Sertraline, citalopram / Fluoxetine	NA	NA	NA	BZD: NA AP: NA Quetiapine: NA
Serbia [40] N-PPD	Yes, if severe after individual risk–benefit assessment	NS	No	Fluoxetine	NA	Paroxetine (data unpublished)	NS	BZD: NA AP: NA Quetiapine: NA
Spain [41] PMH-S	Yes, if severe after individual risk–benefit assessment	NS	NS	Nortriptyline, sertraline, paroxetine / Citalopram, fluoxetine	NA	NA	NS	BZD: NA AP: NA Quetiapine: NA
United Kingdom [44,45] PMH-S	Yes, particularly for moderate-to-severe depression after discussing with the woman of the risk–benefit assessment of AD; drug choice based on lowest risk, monotherapy if possible and at the lowest effective dose.	Consider risks and benefits of BF, which should generally be encouraged, but monitor baby for any adverse effects.	Option to be discussed with the woman, but aim is to expose the breastfed infant to as few drugs as possible.	Unspecified, choice based on prior drug response and its safety profile in breastfeeding.	5.5–12.9 [29,46]	SSRI [49]	Yes, but do not offer BZD except for the short-term treatment of severe anxiety. BZD best avoided in BF if possible; use drug with shortest half-life.	BZD: NA AP: 0.4 [46] Quetiapine: NA

NA = not available; NS = not specified; AD = antidepressant; BF = breastfeeding; BZD = benzodiazepines; AP = antipsychotics; SAH = sedative antihistamines; PMH-S = peripartum mental health specific.

## Data Availability

All data are available online in Medline (via PubMed).

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
