# Peer review of "Treatment of Peripartum Depression with Antidepressants and Other Psychotropic Medications: A Synthesis of Clinical Practice Guidelines in Europe"

_ijerph, 2022, doi:10.3390/ijerph19041973_

Round 1
Reviewer 1 Report
The study was well conducted. The authors included paper based on specific inclusion criteria and used appropriate search strategy. The sources used to search for studies was adequate, criteria to apprise the studies was appropriate. The results are relevant with the study objectives. Moreover, the authors also recommend the practice by the supported data and update the CPG in Europe.
However, the review question is not clear. The authors should state it by describing state of the art regarding to the study. The authors also mention that data abstraction was done by one author (AL) and thereafter quality-checked. It is better if the authors describe the quality-check personnels since the critical appraisal should be conducted by two or more reviewers independently.Author Response
We thank the Editors and the Reviewers for the opportunity to revise our manuscript and for the valuable feedback provided. Please find our replies to each specific comment below.
Comment 1: The study was well conducted. The authors included paper based on specific inclusion criteria and used appropriate search strategy. The sources used to search for studies was adequate, criteria to apprise the studies was appropriate. The results are relevant with the study objectives. Moreover, the authors also recommend the practice by the supported data and update the CPG in Europe.
However, the review question is not clear. The authors should state it by describing state of the art regarding to the study.
Reply 1: Thank you for the positive feedback on our work. We have now provided a more comprehensive background and rationale for conducting this work. The amended text in the Introduction now reads: “In 2018, one systematic review of international CPGs found that only four countries recommend continuation into pregnancy of a pre-existing antidepressant treatment [14]. This prior work extracted only recommendations from CPGs adhering to quality criteria of the Appraisal of Guidelines for Research and Evaluation (AGREE) instrument. Thus, there are still knowledge gaps on current clinical practices from CPGs not meeting such quality criteria. Further, the extent to which national CPGs are followed in relation to antidepressant and other psychotropic prescribing, remains un-known.”
Comment 2: The authors also mention that data abstraction was done by one author (AL) and thereafter quality-checked. It is better if the authors describe the quality-check personnels since the critical appraisal should be conducted by two or more reviewers independently.
Reply 2: We have rephrased this passage of the methods. More than one author carried out data abstraction depending on the language of interest of the clinical practice guideline. AL did abstraction of data for more than one language. Thereafter, a different author checked the content of the data abstracted. Because the two tasks were done by different authors depending on the language of interest, we have rephrased the text as follows: “Data abstraction was done by one author depending on the relevant language, and thereafter quality-checked by another author.”
Reviewer 2 Report
I am reviewing “Treatment of Peripartum Depression with Antidepressant and Other Psychotropic Mediations: A Synthesis of Clinical Practice Guidelines in Europe”. Although the paper presents an overwhelming amount of information, qualitative studies tend to present that way because aggregates cannot be calculated. The main point of the paper is a good one, imploring that Europe needs Clinical Practice Guidelines for distribution of medication for peripartum depression and the anxiety that accompanies it. I make some suggestions to improve the prose.
Line 30 should say “This study examined 1) the availability”. Line 32 should say “Europe, and 2) antidepressant”. Line 33 says “in Medline” but that phrasing is weird. Maybe say “using”. Line 36 should say “14 of them for full-text”.
Line 50 should say “depression, which is depression”. Line 51 should say “using a broad definition”. Line 52 should say “reported” or “experienced” instead of “had”. I do not know what line 54 is saying. Line 55 should not start “Latest research”. Line 57 should put commas around “therefore”. Line 61 should say “long-term negative outcomes”. The authors should delete the comma after “pregnancy”. Line 71 should say “those individuals who chose to experience low”. Line 78 should say “countries have not established CPGs”. Line 83 should say “and evaluated their content”. Line 86 should say “light” instead of “lights”.
Line 100-106 creates an extremely long sentence. Line 107 should say “No exclusion criteria were employed based on language”. Line 113 should say “adults who did not”. Line 115 should say “recommendations”. Line 117 should say “screenings”. Line 118 should say “agreed upon by authors”. Line 119 should put commas around “thereafter”. Lines 119-125 are too long for a single sentence. Line 132 should say “10-to-15 years”. Line 133 should say “as to the way”. Line 136 should say “The outcome criteria were”. Line 139 should say “this review evaluated existing”. Lines 139-141 created a single-sentence paragraph.
Line 154 say “in place, which included”. Line 156 should say “Belgium and Sweden use protocol…based on international…(NICE). However, pharmacotherapy”. Line 160 should say “duplicates of the ones obtained”. Lines 165-167 are confusing. Line 173 should say “The filled prescription…included a decrease”. Line 176 should say “Europe, no such …data were available.” Line 186 perhaps should say “missing for”. Line 194 should say “whereas” rather than “while”. Line 200 should say “amongst the most commonly”.
Line 217 should say “breastfeeding, whereas the UK and Denmark advise to closely”. Line 220 should say “gives a detailed”. Line 221 needs parentheses around some of the text. Line 248 should say “quetiapine specifically are lacking”.
Line 276 should say “This review”. Lines 290-292 should be contained by parentheses with the example. Line 292 should say “Finally” instead of “Lastly”. Line 305 should say “antidepressant effectiveness”. Lines 307-308 should put commas around “however”. The comma after discontinuation should be removed. Line 209 should put a comma after “changes”. Line 311 should put commas around “to date”. Line 317 should say “women, demonstrating the need for larger trials, which”. Line 319 should say “had improved”. Line 322 should say “treatment for”. Lines 323-324 should say “decisions involving CPGs at an international level.”
Line 326 should say “depression to guide”. Line 326 says “However, this” but this what? Lines 331-332 should say “antidepressant outcomes…medication involving the woman’s treatment preference and her design”. Line 333 should say “depressant with the lowest…for breastfed children”. Line 334 should put a comma after “concentration”. Line 336 should say “infants. However, caution”. Line 345 should say “choice was derived”. Line 349 should say “pregnancy, yet only 13.9% of them adhered”. Line 350 should say “are, therefore, necessary to facilitate”.
Line 355 should say “and, when present, it”. Line 358 should say “recommendations for this problem are needed for multiple”. Line 359 should say “symptoms despite”. Line 360 should say “guidance to treat them …anxiety is a prominent symptom of severe”. Lines 363-364 should say “to date, evidence does not exist to help make recommendations for the perinatal population”. Line 370 should say “restriction, which enabled”. Line 372 should remove the comma after “psychology”. Line 373 should say “Europe, which are unlikely to publish national CPGs.” Line 374 should say “consensus statements”. Line 375 should say “these items”. Line 378 should say “are necessary to accurately measure prescribers’ adherence”. Line 387 should say “review expresses the urgent”.
Author Response
We thank the Reviewer for the opportunity to revise our manuscript and for the valuable feedback provided.
To facilitate readability, we have numbered all comments. Our replies to each individual comment are provided below . All changes to the manuscript have been done using track changes in Word.
Comment 1: I am reviewing “Treatment of Peripartum Depression with Antidepressant and Other Psychotropic Mediations: A Synthesis of Clinical Practice Guidelines in Europe”. Although the paper presents an overwhelming amount of information, qualitative studies tend to present that way because aggregates cannot be calculated. The main point of the paper is a good one, imploring that Europe needs Clinical Practice Guidelines for distribution of medication for peripartum depression and the anxiety that accompanies it. I make some suggestions to improve the prose.
Line 30 should say “This study examined 1) the availability”. Line 32 should say “Europe, and 2) antidepressant”. Line 33 says “in Medline” but that phrasing is weird. Maybe say “using”. Line 36 should say “14 of them for full-text”.
Reply 1: We thank the Reviewer for the important feedback provided. We have corrected the lines 30-36 as recommended.
Comment 2: Line 50 should say “depression, which is depression”. Line 51 should say “using a broad definition”. Line 52 should say “reported” or “experienced” instead of “had”. I do not know what line 54 is saying. Line 55 should not start “Latest research”. Line 57 should put commas around “therefore”. Line 61 should say “long-term negative outcomes”. The authors should delete the comma after “pregnancy”. Line 71 should say “those individuals who chose to experience low”. Line 78 should say “countries have not established CPGs”. Line 83 should say “and evaluated their content”. Line 86 should say “light” instead of “lights”.
Reply 2: We have corrected the text as recommended. In line 54, the revised text now reads “In many cases, depression concur with anxiety, and this adds a substantial mental health burden to the woman”. Line 55 now begins with “One recent study”.
Comment 3: Line 100-106 creates an extremely long sentence. Line 107 should say “No exclusion criteria were employed based on language”. Line 113 should say “adults who did not”. Line 115 should say “recommendations”. Line 117 should say “screenings”. Line 118 should say “agreed upon by authors”. Line 119 should put commas around “thereafter”. Lines 119-125 are too long for a single sentence. Line 132 should say “10-to-15 years”. Line 133 should say “as to the way”. Line 136 should say “The outcome criteria were”. Line 139 should say “this review evaluated existing”. Lines 139-141 created a single-sentence paragraph.
Reply 3: We have revised and corrected the text as recommended. Lines 100-106 in the original submission have been re-structured to facilitate readability.
Comment 4: Line 154 say “in place, which included”. Line 156 should say “Belgium and Sweden use protocol…based on international…(NICE). However, pharmacotherapy”. Line 160 should say “duplicates of the ones obtained”. Lines 165-167 are confusing. Line 173 should say “The filled prescription…included a decrease”. Line 176 should say “Europe, no such …data were available.” Line 186 perhaps should say “missing for”. Line 194 should say “whereas” rather than “while”. Line 200 should say “amongst the most commonly”.
Reply 4: We have revised and corrected the text as recommended.
Comment 5: Line 217 should say “breastfeeding, whereas the UK and Denmark advise to closely”. Line 220 should say “gives a detailed”. Line 221 needs parentheses around some of the text. Line 248 should say “quetiapine specifically are lacking”.
Line 276 should say “This review”. Lines 290-292 should be contained by parentheses with the example. Line 292 should say “Finally” instead of “Lastly”. Line 305 should say “antidepressant effectiveness”. Lines 307-308 should put commas around “however”. The comma after discontinuation should be removed. Line 209 should put a comma after “changes”. Line 311 should put commas around “to date”. Line 317 should say “women, demonstrating the need for larger trials, which”. Line 319 should say “had improved”. Line 322 should say “treatment for”. Lines 323-324 should say “decisions involving CPGs at an international level.”
Reply 5: We have revised and corrected the text as recommended.
Comment 6: Line 326 should say “depression to guide”. Line 326 says “However, this” but this what? Lines 331-332 should say “antidepressant outcomes…medication involving the woman’s treatment preference and her design”. Line 333 should say “depressant with the lowest…for breastfed children”. Line 334 should put a comma after “concentration”. Line 336 should say “infants. However, caution”. Line 345 should say “choice was derived”. Line 349 should say “pregnancy, yet only 13.9% of them adhered”. Line 350 should say “are, therefore, necessary to facilitate”.
Reply 6: We have revised and corrected the text as recommended.
Comment 7: Line 355 should say “and, when present, it”. Line 358 should say “recommendations for this problem are needed for multiple”. Line 359 should say “symptoms despite”. Line 360 should say “guidance to treat them …anxiety is a prominent symptom of severe”. Lines 363-364 should say “to date, evidence does not exist to help make recommendations for the perinatal population”. Line 370 should say “restriction, which enabled”. Line 372 should remove the comma after “psychology”. Line 373 should say “Europe, which are unlikely to publish national CPGs.” Line 374 should say “consensus statements”. Line 375 should say “these items”. Line 378 should say “are necessary to accurately measure prescribers’ adherence”. Line 387 should say “review expresses the urgent”.
Reply 7: We have revised and corrected the text as recommended.
Reviewer 3 Report
First of all, I would like to congratulate the authors for the very interesting and useful review on "Treatment of peripartum depression with antidepressant and other psychotropic medications: a synthesis of clinical practice guidelines in Europe".
Below are my comments and suggestions for the authors
-Brexanol is a very promising drug, but it has difficulty in administration (long duration of administration), something that I think is useful for the authors to mention.
-Newborns of mothers on antidepressant treatment need special monitoring in the first 24 hours. I would ask the authors to explain if there are such instructions in the protocols.
-Finally I believe it is not against the purpose of the review, a short, paragraph, on electroconvulsive therapy for pregnant women with severe depression.
Author Response
We thank the Reviewer for the important feedback provided. Please find below our replies to each comment.
Comment 1: First of all, I would like to congratulate the authors for the very interesting and useful review on "Treatment of peripartum depression with antidepressant and other psychotropic medications: a synthesis of clinical practice guidelines in Europe".
Below are my comments and suggestions for the authors
-Brexanol is a very promising drug, but it has difficulty in administration (long duration of administration), something that I think is useful for the authors to mention.
Reply 1: Thank you for raising this point. We agree with the Reviewer, and amended the text as follows.” This new drug constitutes an important therapeutic option for women with severe postpartum depression, but its difficult administration in terms of duration and form (i.e., intravenously) may limit its usage.”
Comment 2: -Newborns of mothers on antidepressant treatment need special monitoring in the first 24 hours. I would ask the authors to explain if there are such instructions in the protocols.
Reply 2: We agree with the Reviewer. We reported this information in Table 2, column with title “AD intake by time of BF”.
Comment 3: -Finally I believe it is not against the purpose of the review, a short, paragraph, on electroconvulsive therapy for pregnant women with severe depression.
Reply 3: The clinical practice guidelines reviewed in this synthesis were specific to pharmacological interventions. Thus, we did not extract data on other treatments including electroconvulsive therapy. However, we agree that more evidence about the effectiveness of pharmacological options versus electroconvulsive therapy in severe cases of depression, is needed. We added the following passage in the discussion: “Similarly, more research is warranted about the comparative effectiveness of different pharmacological interventions versus other therapeutic options, for instance electroconvulsive therapy, for treatment of severe perinatal depression”.